# UNDERSENSITIVITY IN NEURAL READING COMPREHENSION

## ABSTRACT

Neural reading comprehension models have recently achieved impressive generalisation results, yet still perform poorly when given adversarially selected input. Most prior work has studied semantically invariant text perturbations which cause a model's prediction to change when it should not. In this work we focus on the complementary problem: excessive prediction undersensitivity where input text is meaningfully changed, and the model's prediction does not change when it should. We formulate a noisy adversarial attack which searches among semantic variations of comprehension questions for which a model still erroneously produces the same answer as the original question – and with an even higher probability. We show that – despite comprising unanswerable questions – *SQuAD2.0* and *NewsQA* models are vulnerable to this attack and commit a substantial fraction of errors on adversarially generated questions. This indicates that current models—even where they can correctly predict the answer—rely on spurious surface patterns and are not necessarily aware of all information provided in a given comprehension question. Developing this further, we experiment with both data augmentation and adversarial training as defence strategies: both are able to substantially decrease a model's vulnerability to undersensitivity attacks on held out evaluation data. Finally, we demonstrate that adversarially robust models generalise better in a biased data setting with a train/evaluation distribution mismatch; they are less prone to overly rely on predictive cues only present in the training set and outperform a conventional model in the biased data setting by up to 11% $F_1$.

## 1 INTRODUCTION

Neural networks can be vulnerable to adversarial input perturbations (Szegedy et al., 2013; Kurakin et al., 2016). In Natural Language Processing (NLP), which operates on discrete symbol sequences, adversarial attacks can take a variety of forms (Ettinger et al., 2017; Alzantot et al., 2018) including character perturbations (Ebrahimi et al., 2018), semantically invariant reformulations (Ribeiro et al., 2018b; Iyyer et al., 2018b) or—specifically in Reading Comprehension (RC)—adversarial text insertions (Jia & Liang, 2017; Wang & Bansal, 2018). A model's inability to handle adversarially chosen input text puts into perspective otherwise impressive generalisation results for in-distribution test sets (Seo et al. (2017); Yu et al. (2018); Devlin et al. (2019); *inter alia*) and constitutes an important caveat to conclusions drawn regarding a model's language understanding abilities.

While semantically invariant text transformations can remarkably alter a model's predictions, the converse problem of model *undersensitivity* is equally troublesome: a model's text input can often be drastically changed in meaning while retaining the original prediction. In particular, previous works (Feng et al., 2018; Ribeiro et al., 2018a; Sugawara et al., 2018) show that even after deletion of all but a small fraction of input words, models often produce the same output. However, such reduced inputs are usually unnatural to a human reader, and it is both unclear what behaviour we should expect from natural language models evaluated on unnatural text, and how to use such unnatural inputs to improve models. In this work, we show that in RC undersensitivity can be probed with automatically generated natural language questions. In turn, we use these to both make RC models more sensitive when they should be, and more robust in the presence of biased training data.

Fig. 1 shows an examples for a BERT LARGE model (Devlin et al., 2019) trained on SQuAD2.0 (Rajpurkar et al., 2018) that is given a text and a comprehension question, i.e. *"What was Fort Caroline*

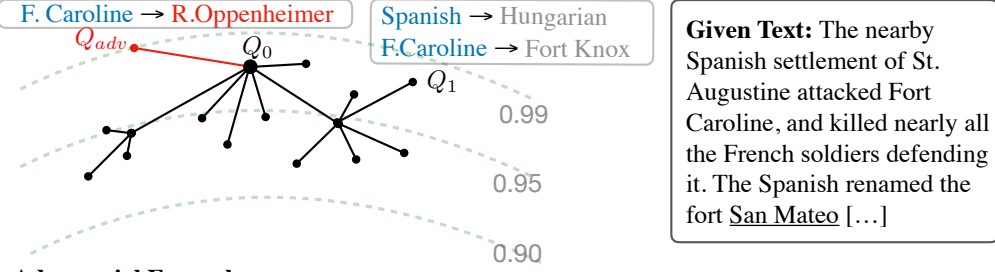

$Q_0$ What was *Fort Caroline* renamed to after the *Spanish* attack? San Mateo (0.98)

**Adversarial Example.**
What was Robert Oppenheimer renamed to after the Spanish attack? San Mateo (0.99) ▲

Figure 1: Method Overview: Adversarial search over semantic variations of RC questions, producing unanswerable questions for which the model retains its predictions with even higher probability.

*renamed to after the Spanish attack?"* which it correctly answers as *"San Mateo"* with 98% confidence. Altering this question, however, can increase model confidence for this same prediction to 99%, even though the new question is unanswerable given the same context. That is, we observe an increase in model probability, despite removing relevant question information and replacing it with irrelevant content.

We formalise the process of finding such questions as an adversarial search in a discrete input space arising from perturbations of the original question. There are two types of discrete perturbations that we consider, based on part-of-speech and named entities, with the aim of obtaining grammatical and semantically consistent alternative questions that do not accidentally have the same correct answer. We find that SQuAD2.0 and NewsQA (Trischler et al., 2017) models can be attacked on a substantial proportion of samples, even with a limited computational adversarial search budget.

The observed undersensitivity correlates negatively with standard performance metrics (EM/$F_1$), suggesting that this phenomenon – where present – is a reflection of a model's lack of question comprehension. When training models to defend against undersensitivity attacks with data augmentation and adversarial training, we observe that they can generalise their robustness to held out evaluation data without sacrificing standard performance. Furthermore, we notice they are also more robust in a learning scenario that has dataset bias with a train/evaluation distribution mismatch, increasing their performance by up to 11%$F_1$. In summary, our contributions are as follows:

- We propose a new type of adversarial attack targeting the undersensitivity of neural RC models, and show that current models are vulnerable to it.

- We compare two defence strategies, data augmentation and adversarial training, and show their effectiveness at reducing undersensitivity errors on held-out data, without sacrificing standard performance.

- We demonstrate that robust models generalise better in a biased data scenario, improving their ability to answer questions with many possible answers when trained on questions with only one.

## 2 RELATED WORK

**Adversarial Attacks in NLP.** Adversarial examples have been studied extensively in NLP – see Zhang et al. (2019) for a recent survey. However, automatically generating adversarial examples in NLP is non-trivial, as the search space is discrete and altering a single word can easily change the semantics of an instance or render it incoherent. Recent work overcomes this issue by focusing on simple semantic-invariant transformations, showing that neural models can be *oversensitive* to such modifications of the inputs. For instance, Ribeiro et al. (2018b) use a set of simple perturbations such as replacing *Who is* with *Who's*. Other semantics-preserving perturbations include typos (Hosseini et al., 2017), the addition of distracting sentences (Jia & Liang, 2017; Wang

& Bansal, 2018), character-level adversarial perturbations (Ebrahimi et al., 2018), and paraphrasing (Iyyer et al., 2018a).

In this work, we instead focus on *undersensitivity* of neural RC models to semantic perturbations of the input. This is related to previous works leveraging domain knowledge for the generation of adversarial examples (Kang et al., 2018; Minervini & Riedel, 2018): our method is based on the idea that modifying, for instance, the named entities involved in a question can completely change its meaning and, as a consequence, the answer to the question should also differ. Our approach does not assume white-box access to the model, as do e.g. Ebrahimi et al. (2018) and Wallace et al. (2019).

**Undersensitivity.** Jacobsen et al. (2019) demonstrated classifier undersensitivity in computer vision, where altered input images can still produce the same prediction scores, achieved using (approximately) invertible networks. Niu & Bansal (2018) investigated over-and undersensitivity in dialogue models and addressed the problem with a max-margin training approach. Ribeiro et al. (2018a) describe a general model diagnosis tool to identify minimal sufficient feature sets that are sufficient for a model to form high-confidence predictions. Feng et al. (2018) showed that it is possible to reduce inputs to minimal input word sequences without changing a model's predictions.

We see our work as a continuation of this line of inquiry, but with a particular focus on undersensitivity in RC. In contrast to Feng et al. (2018), we consider concrete alternative questions, rather than arbitrarily reduced input word sequences. We furthermore address the observed undersensitivity using dedicated training objectives, in contrast to Feng et al. (2018) and Ribeiro et al. (2018a) who simply highlight it.

Finally, one of the baseline methods we later test for defence against under-sensitivity attacks is a form of data augmentation that has similarly been used for de-biasing NLP models (Zhao et al., 2018; Lu et al., 2018).

**Unanswerable Questions in Reading Comprehension.** Following Jia & Liang (2017)'s publication of adversarial attacks on the SQuAD1.1 dataset, Rajpurkar et al. (2018) proposed the SQuAD2.0 dataset, which includes over 43,000 human-curated unanswerable questions. A second dataset with unanswerable question is NewsQA (Trischler et al., 2017), comprising questions about news texts. Training on these datasets should result in models with an ability to tell whether questions are answerable or not; we will see, however, that this does not extend to adversarially chosen unanswerable questions in our undersensitivity attacks. Hu et al. (2019) address unanswerability of questions from a given text using additional verification steps. Other approaches have shown the benefit of synthetic data to improve performance in SQuAD2.0 (Zhu et al., 2019; Alberti et al., 2019).

We operate on the same underlying research premise that the ability to handle unanswerable questions is an important part of improving text comprehension models. In contrast to prior work, we demonstrate that despite improving performance on test sets that include unanswerable questions, the problem persists when adversarially choosing from a larger space of questions.

## 3 METHODOLOGY

**Problem Overview.** Consider a discriminative model $f_\theta$, parameterised by a collection of dense vectors $\theta$, which transforms an input $x$ into a prediction $\hat{y} = f_\theta(x)$. In our task, $x = (t, q)$ is a given text $t$ paired with a question $q$ about this text. The label $y$ is the answer to $q$ where it exists, or a *NoAnswer* label where it cannot be answered.[1]

In a text comprehension setting with a very large set of possible answers, predictions $\hat{y}$ should be *specific* to $x$, i.e. not the model prediction for *arbitrary* inputs. And indeed, randomly choosing a different input $(t', q')$ is usually associated with a change of the model prediction $\hat{y}$. However, there exist many examples where the prediction erroneously remains stable; the goal of the attack formulated here is to find such cases. Concretely, given a computational search budget, the goal is to discover inputs $x'$, for which the model still erroneously predicts $f_\theta(x') = f_\theta(x)$, even though $x'$ is not answerable from the text.

---

[1] Unanswerable questions are part of, e.g. the SQuAD2.0 and NewsQA datasets, but not SQuAD1.1.

**Input Perturbation Spaces.** Identifying suitable candidates for $\boldsymbol{x}'$ can be achieved in manifold ways. A simple option is to search among a large question collection, but we find this approach to only rarely be successful; an example is shown in Table 8, Appendix C. Generating $\boldsymbol{x}'$, on the other hand, is prone to result in ungrammatical or otherwise infeasible text. Instead, we consider a perturbation space $\mathcal{X}_{\mathcal{T}}(\boldsymbol{x})$ spanned by perturbing original inputs $\boldsymbol{x}$ using a perturbation function family $\mathcal{T}$:

$$\mathcal{X}_{\mathcal{T}}(\boldsymbol{x}) = \{T_i(\boldsymbol{x}) \mid T_i \in \mathcal{T}\} \tag{1}$$

This space $\mathcal{X}_{\mathcal{T}}(\boldsymbol{x})$ contains alternative model inputs derived from $\boldsymbol{x}$. Ideally the transformation function family $\mathcal{T}$ is chosen such that the correct label of these new inputs is changed: for $\boldsymbol{x}' \in \mathcal{X}_{\mathcal{T}}(\boldsymbol{x}) : y(\boldsymbol{x}') \neq y(\boldsymbol{x})$. We will later search in $\mathcal{X}_{\mathcal{T}}(\boldsymbol{x})$ to find inputs $\boldsymbol{x}'$ which erroneously retain the same prediction as $\boldsymbol{x}$: $\hat{y}(\boldsymbol{x}) = \hat{y}(\boldsymbol{x}')$.

**Part-of-Speech (PoS) Perturbations.** We first consider the perturbation space $\mathcal{X}_{\mathcal{T}_P}(\boldsymbol{x})$ generated by PoS perturbations $\mathcal{T}_P$ of the original question: we swap individual tokens with other, PoS-consistent alternative tokens, where we draw from large collections of tokens of the same PoS types. For example, we might alter the question *Who patronized the monks in Italy?* to *Who betrayed the monks in Italy?* by replacing the past tense verb *patronized* with *betrayed*. There is no guarantee that the altered question will require a different answer (e.g. due to synonyms). Even more so – there might be type clashes or other semantic inconsistencies (e.g. *Who built the monks in Italy?*). We perform a qualitative analysis to investigate the extent of this problem and find that, while a valid concern, for the majority of attackable samples there exist attacks based on correct well-formed questions (see Section 5).

**Named Entity Perturbations.** The space $\mathcal{X}_{\mathcal{T}_E}(\boldsymbol{x})$ generated by the transformation family $\mathcal{T}_E$ is created by substituting mentions of named entities in the question with different type-consistent named entities, derived from a large collection $E$. For example, a comprehension question *Who patronized the monks in Italy?* could be altered to *Who patronized the monks in Las Vegas?*, replacing the geopolitical entity *Italy* with *Las Vegas*, chosen from $E$. Altering named entities often changes the specifics of the question and poses different requirements to the answer, which are unlikely to be satisfied from what is stated in the given text $t$, given the broad nature of possible entities in $E$. While it is not guaranteed that perturbed questions are in fact unanswerable or require a different answer, we will find in a following qualitative analysis that in the large majority of cases they do.

**Undersensitivity Attacks.** Thus far we have described different methods of perturbing questions. We will search in the resulting perturbation spaces $\mathcal{X}_{\mathcal{T}_P}(\boldsymbol{x})$ and $\mathcal{X}_{\mathcal{T}_E}(\boldsymbol{x})$ for inputs $\boldsymbol{x}'$ for which the model prediction remains constant. However, we pose a slightly stronger requirement: $f_\theta$ should assign an even higher probability to the same prediction $\hat{y}(\boldsymbol{x}) = \hat{y}(\boldsymbol{x}')$ than for the original input:

$$P(\hat{y} \mid \boldsymbol{x}') > P(\hat{y} \mid \boldsymbol{x}) \tag{2}$$

Note that this is a conservative choice, guaranteed to preserve the prediction. To summarise, we are searching in a perturbation space for altered questions which result in a higher model probability to the same answer as the original input question. If we have found such an altered question that satisfies the inequality (2), we have identified a successful adversarial attack, which we will refer to as an *undersensitivity attack*.

**Adversarial Search in Perturbation Space.** In its simplest form, a search for an adversarial attack in the previously defined attack spaces amounts to a search over a list of single lexical alterations for the maximum (or any) higher prediction probability. We can however recur the replacement procedure multiple times, arriving at texts with potentially larger lexical distance to the original question. For example, in two iterations of PoS-consistent lexical replacement, we can alter *Who was the duke in the battle of Hastings?* to inputs like *Who was the duke in the expedition of Roger?*

The space of possibilities with increasing distance grows combinatorially, and with increasing perturbation radius it becomes computationally infeasible to comprehensively cover the full perturbation spaces arising from iterated substitutions. To address this, we follow Feng et al. (2018) and apply beam search to narrow the search space, and seek to maximise the difference

$$\Delta = P(\hat{y} \mid \boldsymbol{x}') - P(\hat{y} \mid \boldsymbol{x}) \tag{3}$$

Beam search is conducted up to a pre-specified maximum perturbation radius $\rho$, but once $\boldsymbol{x}'$ with $\Delta > 0$ has been found, we stop the search.

**Relation to Attacks in Prior Work.** Note that this type of attack stands in contrast to other attacks based on small, *semantically invariant* input perturbations (Belinkov & Bisk, 2018; Ebrahimi et al., 2018; Ribeiro et al., 2018b) which investigate oversensitivity problems. Such semantic *invariance* comes with stronger requirements and relies on synonym dictionaries (Ebrahimi et al., 2018) or paraphrases harvested from back-translation (Iyyer et al., 2018b), which are both incomplete and noisy. Our attack is instead focused on *undersensitivity*, i.e. where the model is stable in its prediction even though it should not be. Consequently the requirements are not as difficult to fulfil when defining perturbation spaces that *alter* the question meaning, and one can rely on sets of entities and PoS examples automatically extracted from a large text collection.

In contrast to prior attacks (Ebrahimi et al., 2018; Wallace et al., 2019), we evaluate each perturbed input with a standard forward pass rather than using a first-order Taylor approximation to estimate the output change induced by a change in input. This is less efficient but exact, and furthermore does not require white-box access to the model and its parameters.

## 4 EXPERIMENTS: MODEL VULNERABILITY

**Training and Dataset Details.** We next conduct experiments using the attacks laid out above to investigate model undersensitivity. We attack the BERT model (Devlin et al., 2019) fine-tuned on SQuAD2.0 (Rajpurkar et al., 2018), and measure to what extent the model exhibits undersensitivity when adversarially choosing input perturbations. Note that SQuAD2.0 per design contains unanswerable questions in both training and evaluation sets; models are thus trained to predict a *NoAnswer* option where a comprehension question cannot be answered.

In a preliminary pilot experiment, we first train a BERT LARGE model on the full training set for 2 epochs, where it reaches, 78.32%EM and 81.44%$F_1$, in close range to results reported by Devlin et al. (2019). We then however choose a different training setup as we would like to conduct adversarial attacks on data entirely inaccessible during training: we split off 5% from the original training set for development purposes and retain the remaining 95% for training, stratified by articles. We use this development data to tune hyperparameters and perform early stopping, evaluated every 5000 steps with batch size 16 and patience 5, and will later tune hyperparameters for defence on it. The original SQuAD2.0 development set is then used as evaluation data, where the model reaches 73.0%EM and 76.5%$F_1$; we will compute the undersensitivity attacks on this entirely held out part of the dataset.

**Attack Details.** To compute the perturbation spaces, we collect large sets of string expressions across Named Entity and PoS types to define the perturbation spaces $\mathcal{T}_E$ and $\mathcal{T}_P$, which we gather from the Wikipedia paragraphs used in the SQuAD2.0 training set, with the pretrained taggers in *spacy*[2], and the Penn Treebank tag set for PoS. This results on average in 5126 different entities per entity type, and 2337 different tokens per PoS tag. When computing PoS perturbations, we found it useful to disregard perturbations of particular PoS types that often led to only minor changes or incorrectly formed expressions, such as punctuation or determiners; more details on the left out tags can be found in Appendix A. As the number of possible perturbations to consider is potentially very large, we limit beam search at each step to a maximum of $\eta$ randomly chosen type-consistent entities from $E$, or tokens from $P$, and re-sample these throughout the search. We use a beam width of $b = 5$, resulting in a bound to the total computation spent on adversarial search of $b \cdot \rho \cdot \eta$ model evaluations per sample, where $\rho$ is the perturbation 'radius' (the maximum search depth).

**Metric: Adversarial Error Rate.** We quantify adversarial vulnerability to the described attacks by measuring the proportion of evaluation samples for which at least one undersensitivity attack is found given a computational search budget, disregarding cases where a model predicts *NoAnswer*.[3]

### 4.1 RESULTS

Fig. 2 depicts adversarial error rates on SQuAD2.0 for both perturbation types across various search budgets. We observe that attacks based on PoS perturbations can already for very small search

---

[2]https://spacy.io
[3]Altering unanswerable samples likely retains their unanswerability.

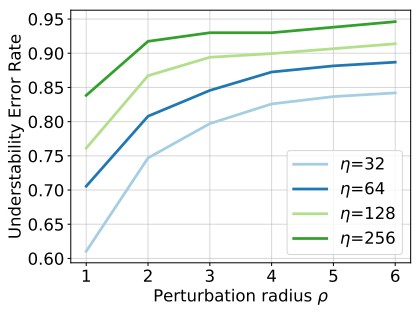
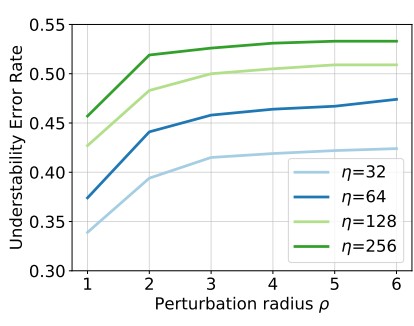

(a) Part of Speech-perturbations

(b) Named Entity-perturbations

Figure 2: BERT LARGE on SQuAD2.0: vulnerability to noisy attacks on held out data for differently sized attack spaces (parameter $\eta$) and different beam search depth (perturbation radius $\rho$).

budgets ($\eta = 32$, $\rho = 1$) reach more than 60% attack success rates, and this number can be raised to 95% with a larger computational budget. For perturbations based on Named Entity substitution, we find overall lower attack success rates, but still find that more than half of the samples can successfully be attacked under the budgets tested. Note that where attacks were found, we observed that there often exist multiple alternatives with higher probability.

These findings demonstrate that BERT is not necessarily specific about the entire contents of a comprehension question given to it, and that even though trained to tell when questions are unanswerable, the model often fails when facing adversarially selected unanswerable questions.

In a side experiment we investigated undersensitivity attacks using Named Entity perturbations on SQuAD1.1, which prove even more vulnerable with an adversarial error rate of 70% already with a budget of $\eta = 32$; $\rho = 1$ (compared to 34% on SQuAD2.0). While this demonstrates that undersensitivity is also an issue for SQuAD1.1, the unanswerable question behaviour is not really well-defined, making results hard to interpret. On the other hand, the notable drop between the datasets demonstrates the effectiveness of the unanswerable questions added during training in SQuAD2.0.

## 5    ANALYSIS AND CHARACTERISTICS OF VULNERABLE SAMPLES

**Qualitative Analysis of the Attacks.**    As pointed out before, the attacks are noisy as the introduced substitutions are by no means guaranteed to result in meaningful and semantically consistent expressions, or require a different answer than the original. To gauge the extent of this we inspect 100 successful attacks conducted at $\rho = 6$ and $\eta = 256$ on SQuAD2.0, both for PoS perturbations and named entity perturbations. We label them as either

1. Having a syntax error (e.g. *What would platform lower if there were fewer people?*). These are mostly due to cascading errors stemming from wrong named entity/PoS tag predictions.

2. Semantically incoherent (e.g. *Who built the monks?*)

3. Questions that require the same correct answer as the original, e.g. due to a paraphrase.

4. Valid attacks: Questions that would either demand a different answer or are unanswerable given the text (e.g. *When did the United States withdraw from the Bretton Woods Accord?* and its perturbed version *When did Tuvalu withdraw from the Bretton Woods Accord?*)

Table 1 shows several example attacks along with their annotations, and in Table 2 the respective proportions are summarised. We observe that a non-negligible portion of questions has some form of syntax error or incoherent semantics, especially for PoS perturbations. Questions with the identical correct answer are comparatively rare. Finally, about half of all attacks in PoS, as well as 84% for named entities are valid questions that should either have a different answer, or be *Unanswerable*.

Overall the named entity perturbations result in much cleaner questions than PoS perturbations, which suffer from semantic inconsistencies in about a quarter of the cases. While these questions have some sort of inconsistency (e.g. *What year did the case go before the supreme court?* vs. a

| Original / Modified Question | Prediction | Annotation | Scores |
|---|---|---|---|
| What city in Victoria is called the cricket ground of [Australia] [the Delhi Metro Rail Corporation Limited]? | Melbourne | valid | 0.63/0.75 |
| What are some of the accepted general principles of [European Union] [Al-Andalus] law? | fundamental rights [...] | valid | 0.59/0.61 |
| What were the [annual] [every year] carriage fees for the channels? | £30m | paraphrase / same answer | 0.95/0.97 |
| What percentage of Victorians are [Christian] [Girlish]? | 61.1% | valid | 0.92/0.93 |
| Which plateau is the left [part] [achievement] of Warsaw on? | moraine | semantic inconsistency | 0.52/0.58 |
| Who leads the [Student] [commissioning] Government? | an Executive Committee | paraphrase / same answer | 0.61/0.65 |

Table 1: Example adversarial questions ([original], [attack]), together with their annotation as either a valid counterexample or other type. Top: Named entity perturbations. Bottom: PoS perturbations.

|  | PoS | NE |
|---|---|---|
| Syntax error | 10% | 6% |
| Semantically incoherent | 24% | 5% |
| Same answer | 15% | 5% |
| Valid attack | 51% | 84% |

Table 2: Analysis of undersensitivity attack samples for both PoS and named entity perturbations.

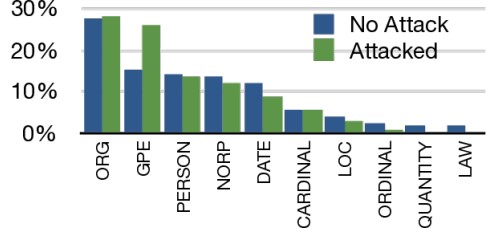

Figure 3: Named entity type characteristics of attackable vs. unattackable samples.

perturbed version *What scorer did the case go before the supreme court?*), it is remarkable that the model assigns higher probabilities to the original answer even when faced with incoherent questions, casting doubt on the extent of question information being used to determine the answer.

Since the named entity-based attacks have a substantially larger fraction of valid, well-posed alternative questions, we will focus our study on these attacks for the remainder of this paper.

## 5.1 CHARACTERISING ATTACKABLE DATA POINTS

We found that models are vulnerable to undersensitivity adversaries, however not all samples were successfully attacked. This raises questions on what distinguishes samples that can and cannot be attacked. We investigate various characteristics, aiming to understand model vulnerability causes.

Questions that can be attacked produce lower original prediction probabilities, with an average of 72.9% compared to 83.8% for unattackable questions. That is, there exists a direct inverse link between a model's original prediction probability and sample vulnerability to an undersensitivity attack. The adversarially chosen questions had an average probability of 78.2%, i.e. a notable gap to the original questions. It is worth noting that search halted once a single question with higher probability was found.

Vulnerable samples are also less likely to be given the correct prediction overall. Concretely, evaluation metrics for vulnerable examples are 56.4%/69.6% EM/$F_1$, compared to 73.0%/76.5% on the whole dataset.

Attackable questions have on average 12.3 tokens, whereas unattackable ones are slightly shorter with on average 11.1 tokens. We considered the distribution of different question types (*What, Who, When, ...*) for both attackable and unattackable samples and did not observe notable differences apart

from the single most frequent question type *What*; it is a lot more prevalent among the unattacked questions (56.4%) than under successfully attacked questions (42.1%). This is by far the most common question type, and furthermore one that is comparatively open-ended and does not prescribe particular type expectations to its answer, as e.g. , a *where* question would require a location. A possible explanation for the prevalence of the *What* questions among defended samples is thus, that the model cannot rely on type constraints alone to arrive at its predictions, and is thus less prone to such exploitation. Section 6.2 will address this in more detail.

Fig. 3 shows a histogram of the 10 most common named entity tags appearing in unattackable samples versus the corresponding fraction of replaced entities in successfully attacked samples. Besides one exception, the distributions are remarkably similar. Undersensitivity can be induced for a variety of entity types used in the perturbation, but in particular questions with geopolitical entities (*GPE*) are error-prone. A possible explanation is observations on (non-contextualised) word embeddings clustering geopolitical entities (e.g. countries) close to one another, thus making them potentially hard to distinguish for a model operating on these embeddings (Mikolov et al., 2013).

## 6 DEFENDING AGAINST UNDERSENSITIVITY ATTACKS

We will now investigate methods for mitigating excessive model undersensitivity. Prior work has considered both data augmentation and adversarial training for more robust models, and we will conduct experiments with both. Adding a robustness objective can negatively impact standard test metrics (Tsipras et al., 2019), and it should be noted that there exists a natural trade-off between performance on one particular test set and performance on a dataset of adversarial inputs. We perform data augmentation and adversarial training by adding a corresponding loss term to the standard log-likelihood training objective:

$$\mathcal{L}^{Total} = \mathcal{L}^{llh}(\Omega) + \lambda \cdot \mathcal{L}^{llh}(\Omega') \tag{4}$$

where $\Omega$ is the standard training data, fit with a discriminative log-likelihood objective, $\Omega'$ either a set of augmentation data points, or of successful adversarial attacks where they exist, and $\lambda > 0$ a hyperparameter. In data augmentation, we randomly sample perturbed input questions, whereas in adversarial training we perform an adversarial search to identify them ($\eta = 32, \rho = 1$). In both cases, alternative data points in $\Omega'$ will be fit to a *NULL* label to represent the *NoAnswer* prediction – which is also fit with a log-likelihood objective. Note that we continuously update $\Omega'$ throughout training to reflect adversarial samples w.r.t the current model.

**Experimental Setup: SQuAD2.0.** We train the BERT LARGE model on SQuAD2.0, tuning the hyperparameter $\lambda \in \{0.0, 0.01, 0.1, 0.25, 0.5, 0.75, 1.0, 2.0\}$ and find $\lambda = 0.25$ to work best for either of the two defence strategies. We tune the threshold for predicting *NoAnswer* based on validation data and report results on the test set (original SQuAD2.0 Dev set). All experiments are run with batch size 16, named entity perturbations for the defence methods, and a relatively cheap adversarial attack with budget $\eta = 32$ and $\rho = 1$. Where no attack is found for a given question we redraw standard samples from the original training data. We evaluate the model on its validation data every 5000 steps (batch size 16) and perform early stopping with a patience of 5.

**Experimental Setup: NewsQA.** Following the experimental protocol for SQuAD, we further test a BERT BASE model on *NewsQA* (Trischler et al., 2017), which – like SQuAD2.0 – contains unanswerable questions. As annotators do often not fully agree on their annotation in NewsQA, we opt for a conservative choice and filter the dataset, such that only samples with the same majority annotation are retained, following the preprocessing pipeline of Talmor & Berant (2019).

**Experimental Outcomes.** Results for these experiments can be found in Table 3 and Table 4 for the two datasets, respectively. First, we observe that both data augmentation and adversarial training substantially reduce the number of undersensitivity errors the model commits, consistently across adversarial search budgets, and consistently across the two datasets. This demonstrates that both training methods are effective defences and can mitigate – but not eliminate – the model's undersensitivity problem. Notably the improved robustness – especially for data augmentation – is possible without sacrificing performance in the overall standard metrics EM and $F_1$, even slight improvements are possible.

| SQuAD2.0 | Undersensitivity Error Rate | | | | HasAns | | NoAns | Overall | |
|---|---|---|---|---|---|---|---|---|---|
| Adv. budget $\eta$ | @32 | @64 | @128 | @256 | EM | $F_1$ | EM/F1 | EM | $F_1$ |
| BERT LARGE | 44.0 | 50.3 | 52.7 | 54.7 | **70.1** | **77.1** | 76.0 | 73.0 | 76.5 |
| + Data Augment. | **4.5** | **9.1** | **11.9** | **18.9** | 66.1 | 72.2 | **80.7** | **73.4** | 76.5 |
| + Adv. Training | 11.0 | 15.9 | 22.8 | 28.3 | 69.0 | 76.4 | 77.1 | 73.0 | **76.7** |

Table 3: Breakdown of undersensitivity error rate overall (lower is better), and standard performance metrics (EM, $F_1$; higher is better) on different subsets of SQuAD2.0 evaluation data, all in [%].

| NewsQA | Undersensitivity Error Rate | | | | HasAns | | NoAns | Overall | |
|---|---|---|---|---|---|---|---|---|---|
| Adv. budget $\eta$ | @32 | @64 | @128 | @256 | EM | $F_1$ | EM/F1 | EM | $F_1$ |
| BERT BASE | 34.2 | 34.7 | 36.4 | 37.3 | **41.6** | 53.1 | 61.6 | 45.7 | 54.8 |
| + Data Augment. | **7.1** | **11.6** | **17.5** | **20.8** | 41.5 | **53.6** | 62.1 | **45.8** | **55.3** |
| + Adv. Training | 20.1 | 24.1 | 26.9 | 29.1 | 39.0 | 50.4 | **67.1** | 44.8 | 53.9 |

Table 4: Breakdown of undersensitivity error rate overall (lower is better), and standard performance metrics (EM, $F_1$; higher is better) on different subsets of NewsQA evaluation data, all in [%].

Second, data augmentation is a more effective defence training strategy than adversarial training. This holds true both in terms of standard and adversarial metrics, and hints potentially at some adversarial overfitting on the training set.

Finally, a closer inspection of how performance changes on answerable (HasAns) vs. unanswerable (NoAns) samples of the datasets reveals that models with modified training objectives show improved performance on unanswerable samples, while sacrificing some performance on answerable samples. [4] This suggests that the trained models – even though similar in standard metrics – evolve on different alleyways during training, and the modified objective prioritise fitting unanswerable questions to a higher degree.

## 6.1 EVALUATION ON HELD-OUT PERTURBATION SPACES

In Tables 3 and 4 results are computed using the same perturbations at training and evaluation time. The perturbation space is relatively large, and questions are about a disjoint set of articles at evaluation time. Nevertheless there is the potential of overfitting to the particular perturbations used during training. To measure the extent to which the defences generalise also to new, held out sets of perturbations, we assembled a new, disjoint perturbation space of identical size per NE tag as those used during training, and evaluate models on attacks with respect to these perturbations. Named entities are chosen from English Wikipedia using the same method as for the training perturbation spaces, and chosen such that they are disjoint from the training perturbation space. We then ran adversarial attacks using these new attack spaces on the previously trained models, and found that both vulnerability rates of the standard model, as well as relative defence success transfer to the new attack spaces. For example, with $\eta = 256$ we observed vulnerability ratios of 51.7%, 20.7%, and 23.8% on SQuAD2 for standard training, data augmentation, and adversarial training, respectively. Detailed results for different values of $\eta$, as well as for NewsQA can be found in Appendix B.

## 6.2 GENERALISATION IN A BIASED DATA SETTING

Datasets for high-level NLP tasks often come with annotation and selection biases; models then learn to exploit shortcut triggers which are dataset- but not task-specific (Jia & Liang, 2017; Gururangan et al., 2018). For example, a model might be confronted with questions/paragraph pairs which only ever contain one type-consistent answer span, e.g. mention one number in a text with a '*How many...?*' question. It is then sufficient to learn to pick out numbers from text to solve the task, irrespective of other information given in the question. Such a model might then have trouble

---

[4]Note that the NoAns prediction threshold is fine-tuned on the respective validation sets.

|  | Person | | Date | | Numerical | |
|---|---|---|---|---|---|---|
|  | EM | $F_1$ | EM | $F_1$ | EM | $F_1$ |
| BERT BASE - w/ data bias | 55.9 | 63.1 | 48.9 | 58.2 | 38.7 | 48.0 |
| + Robust Training | **59.1** | **66.6** | **58.4** | **65.6** | **48.7** | **58.9** |
| BERT BASE - w/o data bias | 69.2 | 78.1 | 73.2 | 81.7 | 69.6 | 80.5 |

Table 5: Robust training leads to improved generalisation under train/test distribution mismatch (data bias, top). Bottom: control experiment without train/test mismatch.

generalising to articles that mention several numbers, as it never learned that it is necessary to take into account other relevant question information that helps determine the correct answer.

We test models trained in such a scenario: a model is trained on SQuAD1.1 questions with paragraphs containing only a single type-consistent answer expression for either a person, date, or numerical answer. At test time, we present it with question/article pairs of the same respective question types, but now there are *multiple* possible type-consistent answers in the paragraph. We obtain such data from Lewis & Fan (2019), who first described this biased data scenario. We use the same training data, but split the test set with a 40/60% split[5] into development and test data.[6] We then test both a vanilla fine-tuned BERT BASE transformer model, and a model trained to be less vulnerable to undersensitivity attacks using data augmentation. Finally, we perform a control experiment, where we join and shuffle all data points from train/dev/test (of each question type, respectively), and split the dataset into new parts of the same size as before, which now follow the same data distribution (w/o data bias setting).

Table 5 shows the results. In this biased data scenario we observe a marked improvement across metrics and answer type categories when a model is trained with unanswerable samples. This demonstrates that the negative training signal stemming from related – but unanswerable – questions counterbalances the signal from answerable questions in such a way, that the model learns to better take into account the relevant information in the question, which allows it to correctly distinguish among several type-consistent answer possibilities in the text, which the standard BERT BASE model does not learn well.

### 6.3 EVALUATION ON ADVERSARIAL SQUAD

We next evaluated BERT LARGE and BERT LARGE + Augmentation Training on ADDSENT and ADDONESENT, which contain adversarially composed samples (Jia & Liang, 2017). Our results, summarised in Table 10 in the Appendix, show that BERT LARGE with robust training improves both EM and F1 on both datasets, boosting F1 by 3.7 and 1.6 points on the two datasets, respectively.

### 6.4 TRANSFERABILITY OF ATTACKS

We trained a RoBERTa model (Liu et al., 2019) on SQuAD2, and conducted undersensitivity attacks ($\rho = 6, \eta = 256$). Attack rates are lower for RoBERTa (34.5%), and when considering only samples where RoBERTa was found vulnerable, BERT also has a vulnerability rate of 90.7%. Concrete adversarial inputs chosen for Roberta transfer when evaluating BERT transfer for 17.5% of samples.

## 7 DISCUSSION AND CONCLUSION

We have investigated a problematic behaviour of RC models – being overly stable in their predictions when given semantically altered questions. This undersensitivity can be drastically reduced with appropriate defences, such as adversarial training, and results in more robust models without sacrificing standard performance. Future work should study in more detail the causes and better defences to model undersensitivity, which we believe provides an alternative viewpoint on evaluating a model's RC capabilities.

---

[5]approximate as we stratify by article

[6]We also include an experiment with the setup used in (Lewis & Fan, 2019), see Appendix E.

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

| SQuAD2.0 | Undersensitivity Error Rate | | | |
|---|---|---|---|---|
| Adv. budget $\eta$ | @32 | @64 | @128 | @256 |
| BERT LARGE | 40.7 | 45.2 | 48.6 | 51.7 |
| + Data Augment. | 4.8 | 7.9 | 11.9 | 20.7 |
| + Adv. Training | 9.2 | 12.2 | 16.5 | 23.8 |

Table 6: Breakdown of undersensitivity error rate on SQuAD2.0 with a held-out attack space (lower is better).

| NewsQA | Undersensitivity Error Rate | | | |
|---|---|---|---|---|
| Adv. budget $\eta$ | @32 | @64 | @128 | @256 |
| BERT BASE | 32.8 | 33.9 | 35.0 | 36.2 |
| + Data Augment. | 3.9 | 6.5 | 11.9 | 17.5 |
| + Adv. Training | 17.6 | 20.7 | 25.4 | 28.5 |

Table 7: Breakdown of undersensitivity error rate on NewsQA with a held-out attack space (lower is better).

## A  APPENDIX: POS PERTURBATION DETAILS.

We exclude these PoS-tags when computing perturbations: *'IN', 'DT', '.', 'VBD', 'VBZ', 'WP', 'WRB', 'WDT', 'CC', 'MD', 'TO'*.

## B  APPENDIX: GENERALISATION TO HELD-OUT PERTURBATIONS

Vulnerability results for new, held-out perturbation spaces, disjoint from those used during training, can be found in Table 6 for SQuAD2, and in 7 for NewsQA.

## C  APPENDIX: ADVERSARIAL EXAMPLE FROM A QUESTION COLLECTION

Searching in a large collection of (mostly unrelated) natural language questions, e.g. among all questions in the SQuAD training set, yields several cases where the prediction of the model increases, compared to the original question, see Table 8 for one example. Such cases are however rare, and we found the yield of this type of search to be very low.

## D  APPENDIX: ATTACK EXAMPLES

Table 11 shows more examples of successful adversarial attacks for NER perturbations on SQuAD2.0.

## E  APPENDIX: BIASED DATA SETUP

For completeness and direct comparability, we also include an experiment with the same data setup chosen in (Lewis & Fan, 2019) (not holding aside a dedicated validation set). Results can be found in Table 9. We again observe improvements in the biased data setting, and the robust model outperforms GQA (Lewis & Fan, 2019) in two of the three subtasks.

## F  APPENDIX: VULNERABILITY ANALYSIS ON NEWSQA

Fig. 4 depicts the vulnerability of a BERT LARGE model on *NewsQA* under attacks using named entity perturbations.

| | |
|---|---|
| **Given Text** | [...] The Normans were famed for their martial spirit and eventually for their Christian piety, becoming exponents of the *Catholic orthodoxy* [...] |
| **Q (orig)** | What religion were the Normans? (78.25%) |
| **Q (adv.)** | IP and AM are most commonly defined by what type of proof system? (83.79%) |

Table 8: Drastic example for lack of specificity: unrelated questions can trigger the same prediction (here: *Catholic orthodoxy*), and even with higher probability.

| | Person | | Date | | Numerical | |
|---|---|---|---|---|---|---|
| | EM | $F_1$ | EM | $F_1$ | EM | $F_1$ |
| GQA (Lewis & Fan, 2019) | 53.1 | 61.9 | 64.7 | 72.5 | **58.5** | **67.6** |
| BERT BASE - w/ data bias | 66.0 | 72.5 | 67.1 | 72.0 | 46.6 | 54.5 |
| + Robust Training | **67.4** | **72.8** | **68.1** | **74.4** | 56.3 | 64.5 |

Table 9: Robust training leads to improved generalisation under train/test distribution mismatch (data bias).

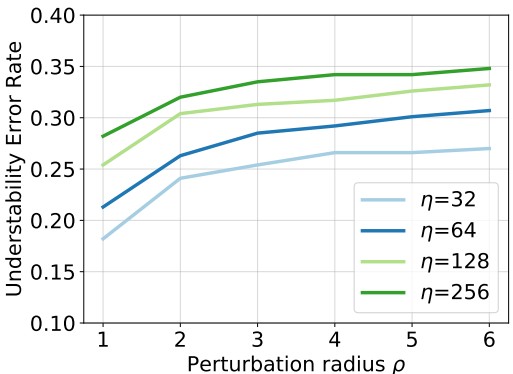

Figure 4: Vulnerability to undersensitivity attacks on NewsQA.

| | ADDSENT | | ADDONESENT | |
|---|---|---|---|---|
| | **EM** | **$F_1$** | **EM** | **$F_1$** |
| LR Baseline | 17.0 | 23.2 | 22.3 | 30.4 |
| Match-LSTM (E) | 24.3 | 34.2 | 34.8 | 41.8 |
| BiDAF (E) | 29.6 | 34.2 | 40.7 | 46.9 |
| SEDT (E) | 30.0 | 35.0 | 40.0 | 46.5 |
| Mnemonic Reader (S) | 39.8 | 46.6 | 48.5 | 56.0 |
| Mnemonic Reader (E) | 40.7 | 46.2 | 48.7 | 55.3 |
| ReasoNet (E) | 34.6 | 39.4 | 43.6 | 49.8 |
| FusioNet (E) | 46.2 | 51.4 | 54.7 | 60.7 |
| GQA | – | 47.3 | – | 57.8 |
| BERT Large | 61.3 | 66.0 | 70.1 | 74.9 |
| BERT Large+NE | **64.0** | **70.3** | **70.2** | **76.5** |

Table 10: Comparison between BERT LARGE and BERT LARGE + Robust Training on two sets of adversarial examples: ADDSENT and ADDONESENT from Jia & Liang (2017) – results from models different from BERT are from Huang et al. (2018), GQA from (Lewis & Fan, 2019). S: single model; E: ensemble model.

| Original / Modified Question | Prediction | Annotation |
|---|---|---|
| What city in Victoria is called the cricket ground of [Australia] [the Delhi Metro Rail Corporation Limited]? | Melbourne | valid |
| What ethnic neighborhood in [Fresno] [Kilbride] had primarily Japanese residents in 1940? | Chinatown | valid |
| What are some of the accepted general principles of [European Union] [Al-Andalus] law? | fundamental rights [...] | valid |
| The [Mitchell Tower] [MIT] is designed to look like what Oxford tower? | Magdalen Tower | valid |
| What were the [annual] [every year] carriage fees for the channels? | £30m | paraphrase / same answer |
| What percentage of Victorians are [Christian] [Girlish]? | 61.1% | valid |
| What does the EU's [legitimacy] [digimon] rest on? | the ultimate authority of [...] | valid |
| What is Jacksonville's hottest recorded [temperature] [atm]? | 104°F | valid |
| Which plateau is the left [part] [achievement] of Warsaw on? | moraine | semantic inconsistency |
| Who leads the [Student] [commisioning] Government? | an Executive Committee | paraphrase / same answer |

Table 11: Example adversarial questions ([original], [attack]), together with their annotation as either a valid counterexample or other type. Top: Named entity perturbations. Bottom: PoS perturbations.

