# OpenReview forum: "Undersensitivity in Neural Reading Comprehension"
_ICLR.cc/2020/Conference — Reject_

### Official Review · AnonReviewer2 · 2019-10-20
**Official Blind Review #2**

**Rating:** 6

**Review:**

This paper studies undersensitivity of the neural models for reading comprehension. First, the neural model is trained on reading comprehension tasks with unanswerable questions. Then, they add perturbations to the input to turn an answerable question into an unanswerable question, using two methods, POS tag based and named entity based. Then, they search for adversarial attacks to find perturbations that the model still predicts the same prediction with even a higher probability. Experiments show that the error rate (attack success rate) is high, over 0.9 with POS tag based method and over 0.5 with named entity based method. Finally, this paper shows data augmentation and adversarial training for this perturbation help the model to be more robust, especially in a biased data scenario.

The contribution of this paper is clear to me: it is one of the first studies which investigates undersensitivity of the model when the input text after the perturbation is complete (e.g. in contrast to Feng et al 2018 and other related work where the perturbation causes the input text to be incomplete).

The weakness of this paper is:
1) the observations are somewhat obvious: it is hard to expect the model to always assign lower probabilities to the original answer when, for example, the named entity in the question is replaced to entities with the same type. Also, I think the observation could be more interesting if the adversarial attack works across different models.
2) Table 2 shows that the perturbation does not always work; especially with POS based method, only half of cases work. How many samples were used for this analysis? Is there a breakdown of the error rate (attack success rate) showing that the rate is still significant for valid perturbations? I think it is significant since perturbations seem to cause invalid attack with a pretty high probability.

Despite the weakness, I think this paper demonstrates comprehensive studies on this focused area and is worth to be published in ICLR overall.


**Experience Assessment:**

I have published one or two papers in this area.

**Review Assessment: Checking Correctness Of Derivations And Theory:**

I assessed the sensibility of the derivations and theory.

**Review Assessment: Checking Correctness Of Experiments:**

I assessed the sensibility of the experiments.

**Review Assessment: Thoroughness In Paper Reading:**

I read the paper at least twice and used my best judgement in assessing the paper.

---

> ### Author Response · Authors · 2019-11-13
> **Response #2**
>
> Thank you for your review.
>
> Yes, one could expect that the model changes its prediction probabilities as entities in the question are exchanged, however models trained on both SQuAD2.0 and NewsQA are expected to detect when a question is unanswerable, as this is explicitly annotated and a requirement to complete the task. This runs contrary to our initial intuition.
> It is also not immediately obvious that models trained with a defence against this problem show substantial improvements in a biased data setting (Section 6.2), or whether sample attackability transfers between models.
>
> Following your suggestion we computed the same attack on the RoBERTa model (Liu et al. 2019, https://arxiv.org/abs/1907.11692), and find a similar picture in terms of model vulnerability. For example, with an attack budget of (rho=6, eta=256) we again find a substantial, but notably lower number of vulnerable samples (34.5% for RoBERTa, compared to 54.7% with BERT). Moreover, we find that vulnerability is transferable between models: those samples vulnerable under RoBERTa have a vulnerability rate of 90.7% on BERT, and 17.5% of concrete attacks transfer (Section 6.4).
> This suggests that besides stronger nominal performance on a variety of tasks, RoBERTa is also more robust than BERT w.r.t. undersensitivity attacks, and that particular undersensitivity blind spots are shared between models despite their significant difference in absolute performance. This might be related to the particular inductive bias of them sharing the same model category, but we leave such an investigation for future work.
>
> Regarding your second comment 2), we had randomly sampled 100 samples of successful attacks for each of the two analyses. As a concrete example, a breakdown can then be computed as follows: with 51% valid PoS attacks, and a vulnerability of 95% (eta=256, rho=6), ~48% of attacks are valid (0.51*0.95=0.4845), which is about half of all samples. Note that there are usually several successful attacks per sample, whereas this analysis only considers a single attack; this number is thus rather a lower bound on the extent of this vulnerability.
>
> Thank you for engaging with our work and your feedback.

---

### Official Review · AnonReviewer3 · 2019-10-23
**Official Blind Review #3**

**Rating:** 3

**Review:**

The paper proposes a framework for evaluating the sensitivity of a QA model to perturbations in the input. The core of the idea is that one can replace content words (i.e. named entities and nouns) in questions in such a way that makes QA models more confident of their original answer (despite, presumably, the question now being unanswerable). Replacements are constructed by mining equivalence classes in Squad data (i.e. all words w/ pos = noun are one set).    Depending on how many such substitutions are searched over (and whether multiple are applied),  one can find at least one such failure in about 50% of cases, on a BERT model trained on Squad2.  The paper also proposes a simple mitigation technique: an objective that modifies a given QA example with all possible substitutions and trains for "no answer" (or alternatively substitutions which break the system).  Results demonstrate that performance on Squad2 is roughly unchanged while the success rate of the attack is significantly decreased.

While the idea of forming such equivalence sets is very interesting, my concern with the paper is both in terms of impact and experimental methodology.

Impact: the method is essentially a data augmentation approach over a fixed list of words. This isn't very different than what was proposed in https://arxiv.org/pdf/1804.06876.pdf and https://arxiv.org/pdf/1807.11714.pdf . While there are some nice nuggets in the analysis, in particular that model confidence is a factor for the attack, I'm not sure anything very novel is being proposed.

Experimental Methodology: Other works in this vein explicitly create a split between counterfactual examples evaluated at train vs at test. The methodology proposed here requires a search where there isn't a clear split between what aspects of the search are allowed at train vs test. In doing counterfactual data augmentation, it is possible the model observes most elements of the search that will be evaluated at test time, making it almost inevitable that the search will be less successful after the model is trained. A simple solution would be splitting the equivalence sets into train/test. I was not able to confirm whether or not this happened from the paper.

That being said, the paper did evaluate on Lewis&Fan( https://openreview.net/pdf?id=Bkx0RjA9tX ) 's bias training simulation, which I appreciate, but I was disappointed that (a) the results from Lewis&Fan were not included for comparison, and when compared the augmentation method proposed here works much worse, in some settings, than generative based training.


**Experience Assessment:**

I have published in this field for several years.

**Review Assessment: Checking Correctness Of Derivations And Theory:**

N/A

**Review Assessment: Checking Correctness Of Experiments:**

I carefully checked the experiments.

**Review Assessment: Thoroughness In Paper Reading:**

I read the paper thoroughly.

---

> ### Author Response · Authors · 2019-11-13
> **Response #3**
>
> Dear Reviewer #3,
>
> Thank you for so thoroughly engaging with our paper and your constructive criticism.
>
> We will address your concerns about i) impact ii) experimental methodology iii) comparison with Lewis and Fan (2018).
>
> i) You are right, data augmentation itself is by no means a new approach. It is a commonly used strategy to defend against adversaries, which is why we investigated it as a baseline for adversarial defence. We don’t see data augmentation itself as a core contribution of our work and have further clarified this in the updated version of the paper, by explicitly relating our work to Zhao et al. (2018) and Lu et al. (2018).
> Instead, we see our main contributions as i) establishing and measuring model undersensitivity as a problem, with concrete strategies for deriving altered questions where the model fails ii) investigating how well-established defence methods (such as augmentation) can mitigate the problem iii) relating model undersensitivity to faulty predictive behaviour, which is improved under the robust model (see the biased data experiment in Section 6.2, and the new experiment on AddSent and on AddOneSent (Section 6.3). We also report several new and interesting observations regarding generalisation to held out perturbations (Section 6.1) the adversarial datasets from Jia et al. 2017 (Section 6.3) and investigate the transfer of attackable samples between models (Section 6.4).
>
> For example, the following attack works for _both_ RoBERTa and BERT:
> Text: “James Hutton is often viewed as the first modern geologist. In 1785 he presented a paper entitled Theory of the Earth to the Royal Society of Edinburgh. [...]”
> Original: “In 1785 James Hutton presented what paper to the Royal Society of Edinburgh?”
> Attack: “In 1785 Jacob Ettlinger presented what paper to the Royal Society of Edinburgh?”
> Prediction: “Theory of the Earth”
>
>
> ii) You raised an excellent point regarding the split between counterfactual examples evaluated during training and evaluation. We had not considered this thus far, and the previous experiments presented in the paper all use the same perturbation space both when computing attacks at (adversarial/augmentation) training time, and when measuring robustness at test time. You are correct, models can then potentially only learn to adapt to the particular perturbation space given during training, while failing to generalise their robustness to a different attack space used at evaluation time.
> To address this concern, we conducted a new set of experiments (Section 6.1), where we test all models on an entirely new perturbation space, entirely disjoint from the one used during training: interestingly, the results are very similar to those observed for the first attack space.
> Concretely, we collected new sets of entities from English Wikipedia articles. Then, we randomly selected exactly as many entities per entity type as used during training (e.g. for the named entity type organisation (ORG) there are 26,014 possible perturbations in the attack space at training time, and the new attack space used in our evaluation would also have 26,014 organisations, and again are disjoint from those previously used). Interestingly we observed that examples prone to be vulnerable to an undersensitivity attack w.r.t one attack space are also prone to an attack in the new space, suggesting that the problem is sample-specific, rather than attack space-specific. Thank you very much for this excellent suggestion. We have updated the paper to include these experiments.
>
> iii) Finally, you are right in your observation in regards to the comparison with Lewis and Fan (2018). We had opted for a different experimental setup than Lewis and Fan (2018) to avoid test leakage, and split off part of the data for validation to avoid tuning models on the same dataset that is also used for evaluation. Adopting the same setting used by Lewis and Fan (2018), we again observe relative improvements (0.3, +2.4, +10.0) F1 for the ‘person’, ‘date’, and ‘numerical’ subtasks, respectively, and Lewis and Fan (2018) is outperformed on two of the three subtasks. For completeness, we have updated the paper and included these results in the Appendix to make our work directly comparable to previous work.
>
> We hope that our response and new experiments fully addresses your concerns and thank you for your feedback that allowed us to considerably improve the experimental rigour of our work.

---

### Official Review · AnonReviewer1 · 2019-10-24
**Official Blind Review #1**

**Rating:** 6

**Review:**

As an extension of recent developments on adversarial attacks and defenses, this paper proposes a simple but effective technique called undersensitivity on machine comprehension task, where the input question is changed but the prediction does not change when it should be. They use two linguistically informed tricks; PoS and NER, to produce the perturbations. In addition to that, several techniques are developed for reducing the adversarial search spaces (Eq 1 and 3) and controlling the level of undersensitivity (Eq 2).

In general, the paper is very well written and clear to read. The formulation of the problem is very straightforward, too. I enjoyed reading the overall paper, especially the experimental results, which provides lots of insights about the techniques. The proposed techniques are simple but they are well-executed in the experiment with reasonable justification. Please find my detailed comments below.


Method.
I appreciate the simplicity of the proposed models with clear motivations. Also, validation of the approaches is well-executed in the experiment.

I like the idea of linguistically-controlled perturbations using PoS and NER. However, there might be many other ways to control it: for example, parsing a sentence using a constituency parser and replacing each phrase with corresponding synonyms/antonyms using WordNet might be interesting. Or, based on the parse, negating the verb might be another way to try. I would expect more linguistically-informed perturbations like these, and I could find some of them from (Kang et al 2018, Ebrahimi et al., 2018). Also, adding a couple of them in the experiment might be interesting to understand the underlying logic of the perturbations.

One major concern of the proposed approach is the sub-optimality by the pre-trained RC model. The undersensitivity (Eq 2) and adversarial search (Eq 3) are calculated by the probability scores predicted by the pre-trained models. This means that producing the new sample x’ is only based on the correctness of the pre-trained model on new samples generated, which sounds to be unreliable. Moreover, using the samples produced by this sub-optimal model may be very limited to produce samples under the sub-optimal space of questions. I wonder how the authors tackle this issue in the experiment.

Experiment
Adversarial attacks should show how an existing system is fragile to be attacked, but at the same time augmenting or adversarially training with them needs to improve its generalization power of the system against the attacks. However, many of the adversarial attack papers mostly focus on the former but not the latter part. In this work, authors showed a result of adversarial training/augmentation but its generalization power on original task (i.e., HasAns case) was not that powerful. The unbiased data setup is interesting but still did not provide any insights about generalization from the adversaries. It would be more convincing to see how this generalization from adversarial attacks can take benefits from bit different tasks such as open-end reading comprehension as a perspective of data augmentation.

I see no comparison with other attacking/defending methods in Tables 3 and 4. Adding the recent models (Ebrahimi et al., 2018, Wallance et al., 2019) may help understand how the proposed models are more effective than other techniques.



**Experience Assessment:**

I have read many papers in this area.

**Review Assessment: Checking Correctness Of Derivations And Theory:**

I assessed the sensibility of the derivations and theory.

**Review Assessment: Checking Correctness Of Experiments:**

I assessed the sensibility of the experiments.

**Review Assessment: Thoroughness In Paper Reading:**

I read the paper at least twice and used my best judgement in assessing the paper.

---

> ### Author Response · Authors · 2019-11-13
> **Response #1**
>
> Dear Reviewer #1
>
> Thank you for your time and comments on our work.
>
> Acting on your suggestions and those from other reviewers, we further investigated the generalisation ability of the robust model. We found that it generalises better than the standard model when tested on the AddSent (66.0 to 70.3 F1) and AddOneSent (74.9 to 76.5 F1) datasets, where it -- to the best of our knowledge -- sets a new state of the art.
> We also investigated transfer of attacks, and found that concrete samples that were attackable under Roberta were also attackable under BERT, and for 17.5% even with the same attack. This indicates that the attacks might be specific to samples rather than particular models. Further details can be found in Sections 6.3 and 6.4.
>
> Yes, you are correct: other types of linguistically informed perturbations are conceivable, especially those with richer (e.g. antonym) annotations in WordNet, or constituent spans. Gathering collections of constituent spans is error-prone, and we would expect there to be a substantial proportion of the perturbed questions that are not well-formed, which we already observed with PoS. One would need to work around this problem, which would require a different approach, but we believe that it would be interesting to see how models behave in this setting. However, we believe that the contributions made in this work are sufficient to warrant publication, and thus leave this analysis for future work.
>
> We would like to try and answer your concern regarding sub-optimality, but are unsure if we fully address your question.
> All adversarial examples x’ are generally computed using fully trained models (after fine-tuning on the QA task), and reflect the model in its fully optimised state.
> There is, however, one exception. For adversarial training (and only in this case), we compute adversarial attacks throughout training. That is, early during adversarial training, when the model has not yet learned to fit the QA task well, it will already be used to compute adversarial attacks x’, and try to improve its adversarial vulnerability w.r.t. this currently sub-optimal model. Throughout training the model then improves on the main QA task, and adversarial examples will be computed w.r.t. this gradually improving model — until convergence. If we misunderstood your question we are more than happy to elaborate on this further.
>
> It is an interesting question how adversarial robustness and standard test metrics interact. Some prior work has pointed out that there is “an inherent tension between the goal of adversarial robustness and that of standard generalization” (“Robustness May Be at Odds with Accuracy” by Tsipras et al. ICLR 2019 https://openreview.net/forum?id=SyxAb30cY7). And indeed, in our case, when we reduce a model’s ability to exploit spurious predictive cues, test set accuracy on the HasAns cases deteriorates, though with a slight overall improvement when unanswerable questions are included in the evaluation.
>
> What is, in our view, interesting, is that by relying less on spurious predictive cues the model behaviour changes qualitatively. Two particular insights we drew from the biased data experiment (Section 6.2) are that: i) The standard model does not reliably learn to form predictions that are specific to the entity in question ii) The robust model partly overcomes this, since becoming more robust correlates with taking into account the specific entities in question.
>
> Finally, thank you for your suggestion of adding alternative baselines for the adversarial attack. Thus far there is relatively little prior work on model undersensitivity. The first of your suggestions, HotFlip (Ebrahimi et al. 2018), describes an oversensitivity attack, whereas our attacks attempt to find cases of undersensitivity. Universal Adversarial Triggers (Wallace et al. 2019) on the other hand append new text to SQuAD examples that then triggers a model to predict a subspan of that new text. While all falling under the umbrella of ‘adversarially chosen examples’, the underlying model failures are very distinct from the undersensitivity problem we investigated, and we do not see any straightforward way in which these can be applied in the context of undersensitivity.

---

### Decision · Program_Chairs · 2019-12-19

**Decision:**

Reject

**Comment:**

The paper investigates the sensitivity of a QA model to perturbations in the input, by replacing content words, such as named entities and nouns, in questions to make the question not answerable by the document. Experimental analysis demonstrates while the original QA performance is not hurt, the models become significantly less vulnerable to such attacks. Reviewers all agree that the paper includes a thorough analysis, at the same time they all suggested extensions to the paper, such as comparison to earlier work, experimental results, which the authors made in the revision. However, reviewers also question the novelty of the approach, given data augmentation methods. Hence, I suggest rejecting the paper.